# Model guidance via explanations turns image classifiers into segmentation models

## Abstract

Heatmaps generated on inputs of image classification networks via explainable AI methods like Grad-CAM and LRP have been observed to resemble segmentations of input images in many cases. Consequently, heatmaps have also been leveraged for achieving weakly supervised segmentation with image-level supervision. On the other hand, losses can be imposed on differentiable heatmaps, which has been shown to serve for (1) improving heatmaps to be more human-interpretable, (2) regularization of networks towards better generalization, (3) training diverse ensembles of networks, and (4) for explicitly ignoring confounding input features. Due to the latter use case, the paradigm of imposing losses on heatmaps is often referred to as "Right for the right reasons". We unify these two lines of research by investigating semi-supervised segmentation as a novel use case for the Right for the Right Reasons paradigm. First, we show formal parallels between differentiable heatmap architectures and standard encoder-decoder architectures for image segmentation. Second, we show that such differentiable heatmap architectures yield competitive results when trained with standard segmentation losses. Third, we show that such architectures allow for training with weak supervision in the form of image-level labels and small numbers of pixel-level labels, outperforming comparable encoder-decoder models. All code available upon publication.

## 1 Introduction

Ongoing advancements in explainable AI methodology aim to interpret model behavior and offer human-readable explanations. In the field of computer vision, methods that explain image classification network predictions in the form of *heatmaps*, i.e., assign relevance to each pixel of an input image, have gained wide popularity: Heatmaps are extensively leveraged to gauge the trustworthiness of classifiers, as they have been shown to detect biases and confounders in input images (Lapuschkin et al., 2019; Anders et al., 2022).

The capacity to detect biases and confounders in input images goes naturally hand in hand with the capacity to locate target objects of the semantic classes for which an image classifier has been trained. This capacity has been shown for many well-known XAI methods (Selvaraju et al., 2017; Bach et al., 2015; Nam et al., 2020; Gur et al., 2021; Achtibat et al., 2023), and has motivated approaches for leveraging heatmaps towards weakly supervised segmentation, i.e., segmentation networks that train on image-level labels as opposed to expensive dense, pixel-level labels. In particular, Grad-CAM based heatmaps (Selvaraju et al., 2017) are a popular ingredient in weakly supervised segmentation (WSSS) models (Li et al., 2018; Lee et al., 2019; Kim et al., 2021; 2022).

While the above line of research exploits heatmaps generated by trained models, an orthogonal line of research has tapped into the potential of heatmaps as a lever to influence model training in the first place (Weber et al., 2023), with the aim to yield models that are "Right for the Right Reasons" (Ross et al., 2017). To this end, imposing losses on differentiable heatmaps has been shown to allow for aligning explanations with human interpretation (Liu & Avci, 2019; Ross et al., 2017; Rieger et al., 2020; Shao et al., 2021; Shakya et al., 2022), enhancing model generalizability (Rao et al., 2023), and ignoring undesired image features towards diverse ensembles (Ross et al., 2017; Teney et al., 2022) and confounder mitigation (Schramowski et al., 2020).

In our paper, we aim to integrate these two lines of research by exploring the potential of the Right for the Right Reasons paradigm for semi-supervised segmentation. To this end:

- We establish formal parallels between differentiable heatmap architectures and conventional encoder-decoder architectures commonly used for image segmentation. In particular, we show that "unrolling" LRP (Montavon et al., 2019) on standard image classification architectures like a ResNet50 yields convolutional encoder-decoder architectures that resemble standard U-Nets (Ronneberger et al., 2015), albeit with weights tied between encoder and decoder and modified, "skipped" activation functions in the decoder.

- We perform a comparative evaluation of unrolled LRP and comparable U-Nets in terms of segmentation accuracy on the *val* set of the PASCAL VOC 2012 segmentation benchmark (Everingham et al., 2010). Our experimental results reveal for multiple standard classification backbones that differentiable heatmap architectures, trained with combined classification- and segmentation loss, can achieve competitive segmentation performance.

- We evaluate semi-supervised training, with pixel-level labels ranging down to one labelled image per class, revealing that in scenarios with few pixel-wise labels, unrolled heatmap architectures outperform comparable standard UNets for segmentation by up to 10% mIoU on PASCAL.

**Relation to previous works.** The work of Li et al. (2018) is closely related to ours in that, to our knowledge, it is the sole previous work that also optimizes heatmaps towards improved segmentation performance. To this end, they optimize low-resolution Grad-CAM based heatmaps by means of specifically designed losses, and use resulting label maps as pseudo-labels for further training and sophisticated post-processing to increase segmentation accuracy. Instead, we optimize full-resolution LRP heatmaps by means of a vanilla segmentation loss, and establish formal parallels of this approach to conventional segmentation architectures rather than aiming at state of the art segmentation accuracy via subsequent heuristic processing. Our results outperform Li et al. (2018) by 9% in a comparable scenario without their CRF post-processing.

The recent work of Rao et al. (2023) is closely related to ours as they optimize heatmaps towards improved localization performance under various loss functions and heatmap-generating methods, including IxG, which is equivalent to LRP-0 (Ancona et al., 2017) as employed in our study. However, they do not aim at a formal analysis of the respective unrolled architectures, whereas we reveal strong formal parallels between unrolled LRP-0 and standard encoder-decoder architectures; Furthermore, their work focuses on localization instead of segmentation, and their losses are specifically designed for optimizing heatmaps whereas we employ standard segmentation loss. To this end, we successfully unroll heatmaps for all class scores in each batch, whereas they restrict their losses to one (randomly selected) class per batch. In a striking difference to Rao et al. (2023), we empirically find that classifier performance does not degrade during training while segmentation performance increases significantly. This holds for all pixel-level supervision regimes we study. To the contrary, their empirical results suggest that increased localization performance has a strong tendency to go hand in hand with decreased classification performance. We discuss the hypothesis that this different behavior may be due to the concordant nature of the classification- and heatmap losses we employ.

Our derivation of unrolled heatmap architectures manifests a special case of double backpropagation on ReLU networks, which has been formally studied before by Etmann (2019) and, in the context of heatmap methods, by Alvi et al. (2018). However, to our knowledge, we are first to report formal parallels of unrolled heatmap architectures to standard segmentation networks, and we showcase their respective potential by empirical results that outperform standard segmentation baselines.

**Limitations.** Our work does not aim at state of the art semi-supervised segmentation: We do not employ any post-processing on optimized heatmaps (Li et al., 2018; Wei et al., 2018; Lee et al., 2019; Luo & Yang, 2020), nor do we leverage any orthogonal avenues towards semi-supervised segmentation, like e.g. entropy minimization (Berthelot et al., 2019), consistency regularization (Pan et al., 2022; Lai et al., 2021; Ouali et al., 2020; Sohn et al., 2020; Du et al., 2022), or contrastive learning (Liu et al., 2022). However, thanks to our standard training objective, our unrolled architecture can be directly plugged into these orthogonal approaches.

## 2 UNROLLED HEATMAP ARCHITECTURES

In this Section, we present formal parallels between differentiable heatmap architectures and classical encoder-decoder architectures for image segmentation. After a brief recap of LRP-$\epsilon$ (Bach et al., 2015), we derive the building blocks of architectures entailed by "unrolling" LRP-0 on standard con-

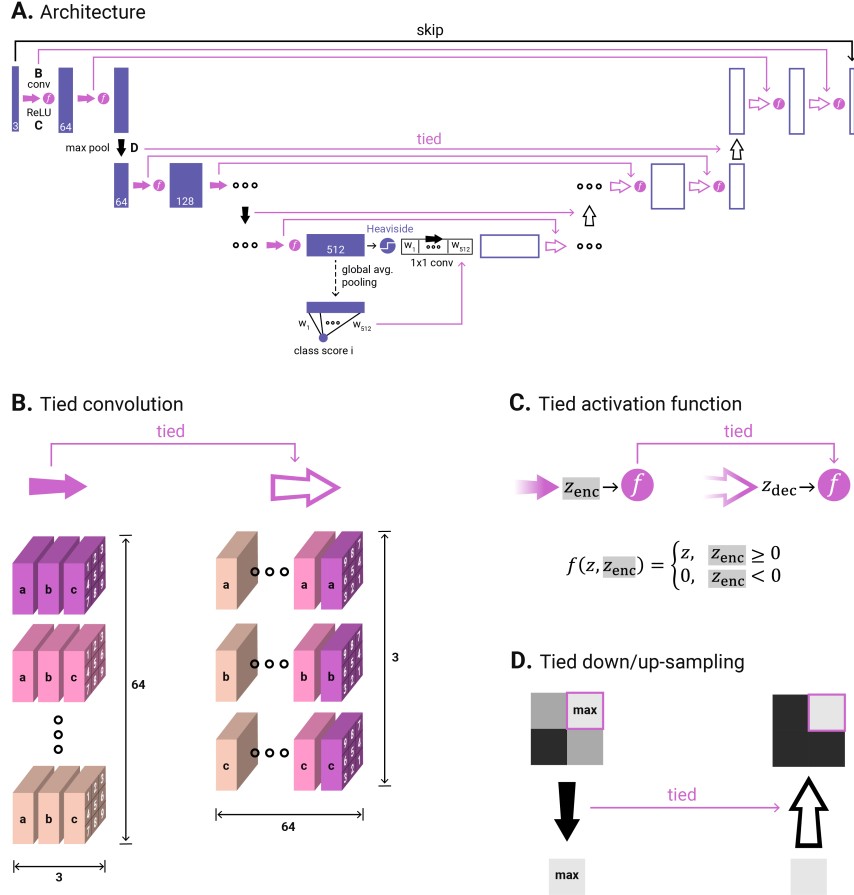

Figure 1: Unrolled LRP-0 for standard convolutional classifiers is an encoder-decoder CNN. (A) Sketch of architectural building blocks. (B) Decoder weights are tied with corresponding encoder layers. (C) Activation functions are skipped from encoder to decoder. (D) Up-convolutions are tied to the respective pooling operation in the encoder. In (A) we sketch the decoder architecture induced by an individual semantic class i; this decoder is replicated for all classes with shared weights, except for the bottleneck 1x1 convolution, which is class-specific.

volutional image classification models, comprising ReLU nonlinearities, max pooling, and global average pooling. The resulting architectural blueprint is sketched out in Figure 1, and forms the basis of our subsequent quantitative analysis of segmentation performance. Last, we assess formal properties of said unrolled architectures.

## 2.1 LRP BASICS

**Linear layer.**    In the following, we denote the input vector of a linear layer $l$ by $a^{(l-1)}$, its weights matrix by $w^{(l)}$, and its output vector (pre-activation) by $z^{(l)}$. Considering a single linear filter in isolation, the relevance LRP in its most basic form (LRP-0) assigns to an input variable $a_i$ equals the variable's contribution to the (scalar) output, namely $R_i = a_i w_i$. Considering a single linear filter in an intermediate layer $l + 1$ of a neural network, assuming the relevance $R_j^{(l+1)}$ of the filter's output $z_j^{(l+1)}$ has already been determined, LRP-0 distributes $R_j^{(l+1)}$ to the filter's inputs $a_i^{(l)}$ proportional to their respective contribution to $z_j^{(l+1)}$, namely as $R_j^{(l+1)} a_i^{(l)} w_{ij}^{(l+1)} / z_j^{(l+1)}$. This ensures preservation of total relevance across layers. Considering a general linear layer $l + 1$ as a whole, input variables accumulate relevances from all linear filters they contribute to by summation. This entails the following recursive definition by which LRP-0 assigns relevances to inputs $a_i^{(l)}$ given

relevances $R_j^{(l+1)}$ of outputs $z_j^{(l+1)}$:

$$R_i^{(l)} = a_i^{(l)} \sum_j \frac{w_{ij}^{(l+1)}}{z_j^{(l+1)} + \epsilon} \cdot R_j^{(l+1)}. \tag{1}$$

The $\epsilon$ in Eq. 1 serves for numerical stability, but may also serve to mitigate noise in the relevances in a variant of LRP termed LRP-$\epsilon$. We focus our analysis on LRP-0, where $\epsilon$ is set to a very small constant that serves solely for numerical stability and can be reasonably ignored for simplified formal analyses. Eq. 1 can be phrased as a three-step algorithm: Applying LRP to any linear layer $l + 1$ amounts to the following sequence of operations applied to $R^{(l+1)}$:

---

**Algorithm 1** LRP through a linear layer $l + 1$

1: Element-wise division by $z^{(l+1)}$ (yielding $q := R^{(l+1)}/z^{(l+1)}$)
2: Weighted summation via $w^{(l+1)}$ (yielding $p_i := \sum_j w_{ij}^{(l+1)} q_j$ for all $i$)
3: Element-wise multiplication by $a^{(l)}$ (yielding $R_i^{(l)} = a_i^{(l)} p_i$ ).

---

**ReLU and pooling.** LRP sets the relevance of the input of a ReLU to be equal to the relevance of its output, i.e., relevances are propagated through ReLUs unchanged. Pooling operations can be treated as their corresponding equivalent convolutions (where for max pooling, individual inputs entail respective individual convolution kernels).

## 2.2 Unrolled LRP Architectures for Convolutional Classifiers

In the following, we derive the building blocks of "unrolled LRP". To this end, we phrase LRP through standard CNN classifier/encoder building blocks as corresponding decoder building blocks.

**Single convolution.** First, we consider a single convolutional filter applied to a stack of activation maps of depth $c_l$ at some layer $l$. A pixel in activation map k contributes to the respective output map in layer $l + 1$ as follows: (1) To a spatial region around its own location that is of the same size as the spatial extent of the convolutional kernel; (2) With weights solely from the k-th slice of the convolutional kernel; (3) To position (dx,dy) relative to its own location with weight at position (-dx,-dy) in the conv kernel (when defining the center point location of the conv kernel to be (0,0)). Thus, the weighted combination entailed by LRP (cf. Step 2 of Alg. 1) through a single convolution of depth $c_l$ (layer $l \rightarrow l + 1$) amounts to the application of $c_l$ convolutions of depth 1 to the relevance map obtained for layer $l+1$. These "unrolled LRP" convolutions are formed by slices of the original convolutional kernels, with weights flipped in all spatial dimensions.

**Convolutional layer.** Next we consider a full convolutional layer, with $c_{l+1}$ convolutional filters yielding $c_{l+1}$ output maps. Here, a pixel in activation map k in layer $l$ contributes to identical spatial regions in all resulting output maps in layer $l + 1$, where weights stem from the k-th slices of each convolutional kernel in the same mirrored manner as described for a single convolutional filter above. This yields $c_{l+1}$ "unrolled LRP" convolutional kernels, where the k-th unrolled kernel is assembled by stacking the k-th slices of all original kernels, mirrored in all spatial directions.

Altogether, the weighted combination entailed by LRP through a convolutional layer with input depth $c_l$ and output depth $c_{l+1}$ amounts to an "unrolled LRP" convolutional layer with input depth $c_{l+1}$ and output depth $c_l$ with tied weights as sketched in Fig. 1 B.

**Consecutive convolutional layers with ReLU activations.** Consider a convolution with ReLU activation followed by another convolution. As LRP passes relevances through ReLUs unaffected, in this case, Alg. 1 is simply applied twice in a row. In the resulting sequence of six operations, step 3 and 4 can be condensed into Element-wise multiplication by

$$a^{(l)}/z^{(l)} = \begin{cases} 1 & z^{(l)} > 0 \\ 0 & \text{otherwise} \end{cases} = H(z^{(l)}),$$

where $H$ denotes the Heaviside function. This entails unrolled LRP "activation functions" between convolutions that do not follow the classical paradigm of a nonlinear function applied to convolution outputs; Instead, the unrolled activation function zeroes out unrolled convolution outputs based on

the output value of the respective original convolution. This can be seen as the same function being applied after original and unrolled convolution, as sketched in Figure 1 C.

Note that pooling layers can be seen as convolutions (with appropriate kernels) and thus merged with a subsequent convolution. Consequently the above unrolled LRP "activation functions" also appear after the last unrolled convolution per downsampling level.

**Final classifier layers.** Consider the final layers of a standard classifier: The final stack of spatial activation maps, which we term $a$ here, are fed through global average pooling, yielding an average pooling vector we term $apv$. $apv$ is then fed through a linear output layer, yielding a vector of class scores. LRP is performed individually per class $j$. To this end, LRP is initialized by setting the relevance of the respective class score equal to its value. Thus the elements of the average pooling vector, $apv_i$, receive relevances $R_i = avp_i w_{ij}$. Considering $avp$ as input, the initial steps of LRP on class score $j$ thus play out as follows:

---

**Algorithm 2** LRP on the class score of an individual class $j$ through final standard classifier layers

1: Element-wise multiplication of $apv$ with the class weights of class $j$ from the last linear layer
2: Element-wise division by $apv$ (i.e., step 1 of Alg. 1 for the average poling layer)
3: Division by the (spatial) number of pixels in $a$, followed by nearest-neighbor upsampling to the spatial extent of $a$ (i.e., step 2 of Alg. 1 for the average poling layer)
4: Element-wise multiplication with $a$ (i.e., step 3 of Alg. 1 for the average poling layer)
5: Element-wise division by $z$ (i.e., step 1 of Alg. 1 for the last conv layer).

---

Step 3 yields a stack of activation maps of spatial extent and depth equal to $a$. Each activation map has a constant value, namely a weight from the output layer class weight vector divided by the (spatial) number of pixels in $a$. Steps 4 and 5 zero out this constant value whereever $z \leq 0$. Thus Algorithm 2 can be condensed into a Heaviside function applied to $z$, followed by a 1x1 convolution, where each of the conv kernels has one non-zero entry, namely one of the class weights. This is sketched out in the bottleneck of the unrolled architecture in Figure 1 A. Note, classifiers without average pooling and/or with multiple fully connected layers before the output layer can be unrolled analogously: Average pooling is equivalent to a fully connected layer with specific weights, and fully connected layers are equivalent to convolutions with suitable kernel size.

To consider not just one class score but the class score vector as a whole, the complete decoder architecture is replicated for each class. The resulting decoder branches are class-specific solely in the first 1x1 convolution with the class-specific weights of the classifier output layer. All other weights are shared between decoder branches.

**Classifier input layer.** LRP through the first conv layer of a classifier is special as multiplication with $a$, which is the input image in this case, does not cancel out by supsequent propagation through another convolution. Thus the last building block of the unrolled architecture is an element-wise multiplication with the input image, depicted as skip connection in Fig. 1 A.

## 2.3 LOSSES AND TRAINING

We turn three-channel heatmaps into single-channel heatmaps by summation as is standard. We then stack together heatmaps from all decoder branches, thus forming class score vectors per pixel, on which we employ standard pixel-wise softmax cross-entropy loss. We complement this standard segmentation loss by standard classification loss on the class score vector in the bottleneck, namely sigmoid cross-entropy per class score, same as during classifier training.

This combination of losses entails interesting training behavior: LRP is designed to satisfy the conservation property, i.e., it aims at conserving total relevance across layers Montavon et al. (2019). Assuming the conservation property holds (and thus ignoring the bias terms for the sake of a simplified argument), the total relevance of a heatmap, i.e., the sum of all pixel relevances, equals the respective class score. For negative classes, i.e., classes not present in an image, this entails the following: Softmax cross-entropy pushes down heatmaps of negative classes at all pixels. Due to conservation, this necessarily also pushes down the respective class score in the bottleneck, which is concordant with the respective classifier loss. An analogous argument holds for a single positive class that covers the whole image: Softmax cross-entropy pushes up the respective heatmap at all

pixels and thus, due to conservation, also the class score in the bottleneck, which is again concordant with the respective classifier loss. For the case of multiple positive classes, which is the most common case due to the abundant background class and our multi-label scenario, the segmentation loss pushes up class-specific foreground pixels in the respective heatmap while all other pixels in this heatmap are pushed down. Consequently, some trade-off needs to be met to not cause the respective class score to decrease. We empirically find that a suitable trade-off is met, as we observe that classification performance does not degrade during training (cf. Sec. 3).

## 2.4 RELATION TO PREVIOUS FORMAL ANALYSES AND STANDARD ARCHITECTURES.

**Previous formal analyses.** Backpropagtion through unrolled LRP-0 architectures as blueprinted in Fig. 1A does not reach the encoder (no matter which heatmap loss is employed): Due to the Heaviside function and its zero local derivative, global derivatives are zeroed out along all paths to the encoder. Consequently, the derivatives of the (intermediate) output directly before the final skip-multiplication with the input w.r.t. the network weights takes a simple form, namely a sum of other weights' products, where the input image determines which products are non-zero. This has been shown before by Ancona et al. (2017) as part of their proof that LRP-0 equals inputs times gradients (IxG) in ReLU networks.

It trivially follows from LRP-0 = IxG that the output of our unrolled architecture, pre skip-multiplication with the input, equals the gradient of the class score w.r.t. the input. Thus our unrolled architecture implements double backpropagation (Drucker & Le Cun, 1992). Double backpropagation has been formally analyzed in-depth by Etmann (2019), who have also derived the respective simplified gradients for ReLU networks, and note that the backward pass is thus shortened.

Our architecture serves as easily accessible special case of what the above more general theory entails: When trained with classification- and heatmap loss, the gradient of the classification loss backpropagates solely through the encoder, while the gradient of the heatmap loss backpropagates solely through the decoder. This can be leveraged for efficient training; It remains unclear whether the property has further consequences, e.g. whether it has a systematic effect on training dynamics; Furthermore, in practice, note that any $\epsilon > 0$ in Eq. 1 smoothes out all Heaviside functions in the positive regime, thus yielding non-zero heatmap loss gradient flow to the encoder during backprop.

**Standard architectures.** On the one hand, unrolled heatmap architectures as described above are related to U-Net architectures Ronneberger et al. (2015), which constitute the state of the art in semantic segmentation in a broad range of applications to date Isensee et al. (2021). A U-Net is related in that it is an encoder-decoder convolutional architecture. It is distinct in that (1) convolution- and pooling operations are not tied between encoder and decoder whereas ours are; (2) it employs standard nonlinearities in the decoder whereas we employ "tied activations"; (3) it employs skip-concatenations of encoder activation maps to the respective decoder layer whereas we only skip-connect activations; and (4) it employs a single decoder for all classes, whereas we employ weight-shared branches per class.

On the other hand, unrolled heatmap architectures are closely related to weight-tied auto-encoders, as first described by Vincent et al. (2010). In particular, Kim & Hwang (2016) have explored weight-tied auto-encoders based on convolutional encoders. The weight-tied architecture they establish equals ours in their tied decoder convolutions and up-sampling operators, yet is distinct in that they employ standard nonlinearities in the decoder, and in that they do append a final, non-tied convolutional layer. Furthermore, like U-Net, they also employ only a single decoder for all classes.

In summary, our unrolled LRP architectures are most constrained to their encoder backbone among all previously described encoder-decoder-style convolutional segmentation architectures.

## 3 UNROLLED HEATMAP ARCHITECTURES FOR SEGMENTATION: RESULTS

**Data and Baselines.** We evaluate the performance of our unrolled architecture on the PASCAL VOC 2012 segmentation benchmark val set. We employ a comparable U-Net, i.e., a U-Net with identical convolutional encoder, as standard segmentation baseline. We evaluate two different classifier backbones (resnet50 and vgg16 with batch norm) in four scenarios with varying levels of pixel-wise supervision, namely with 1, 5, 25, and all available labelled images per class. This amounts to

| Method | 1.4 % (20) | 6.8 % (100) | 34.2 % (500) | Full(1464) |
|---|---|---|---|---|
| *Architecture*: vgg16 backbone for encoder | | | | |
| UNet | 20.38 | 40.50 | 54.60 | 60.53 |
| unrolled LRP (ours) | **33.92** | **49.68** | **59.96** | **63.85** |
| *Architecture*: ResNet50 backbone for encoder | | | | |
| UNet | 25.89 | 43.40 | 55.10 | 60.07 |
| multi-task UNet | 25.06 | 39.77 | 55.13 | 60.37 |
| unrolled LRP (ours) | **39.80** | **50.29** | **58.30** | **61.50** |

Table 1: Segmentation mIOU on PASCAL VOC 2012 *val* for vgg16 and ResNet50 backbones, models, and supervision scenarios, ranging from 20 to 1464 labelled images. We report average mIoU over three independent training runs for all models.

| Backbone | 0 | 1.4 % (20) | 6.8 % (100) | 34.2 % (500) | Full(1464) |
|---|---|---|---|---|---|
| vgg16bn | 81.33 | 80.73 | 81.99 | 81.86 | **82.52** |
| resnet50 | 81.62 | 80.35 | 80.80 | **82.21** | 81.49 |

Table 2: Classification F1 score of our unrolled LRP models on PASCAL VOC 2012 *val* after pre-training (column "0") and after segmentation training across supervision scenarios.

respective totals of 20, 100, 500, and 1464 labelled images. We employ standard augmentations, as well as standard cropping to fit the classifiers' required input size. We initialize the classifier backbones of all architectures to their respective ImageNet pre-trained and PASCAL fine-tuned weights.

To give our UNet baseline the chance to leverage the 15,676 available image-level labels not just via pre-training but also during segmentation training, we include a modified version of the resnet50-encoder variant as additional baseline, where we extend a comparable classifier branch, i.e., simple average pooling followed by a linear output layer, from the UNet bottleneck. We train this "multi-task" UNet with same combined classification- and segmentation loss as our unrolled architecture.

We train all models with batch size 10, AdamW with learning rate $1e-5$. We give equal weight (=1) to classification- and segmentation loss in our unrolled architectures and multi-task UNet baseline.

To cover different semi-supervised scenarios we sample 20, 100, 500 and 1464 labeled images from the training set, respectively. For the scenarios with 20 and 100 labelled images we sample (randomly, with fixed seed for all experiments) maximally uniform distributed classes. For the 20 labeled images case, this ensures that each class is sampled at least once. However, due to the strong class imbalance within PASCAL, when dealing with larger image sets, sampling with a uniform distribution is no longer possible; Thus we relax the strategy for $\geq 500$ labeled images. Here, we sample with the average of a uniform distribution and the class distribution in the training set.

**Results.** Table 1 reports results in terms of segmentation mIoU averaged over three independent training runs per model. Our models outperform comparable UNets in all supervision scenarios, where the margin increases drastically with decreasing pixel-level supervision. At the same time, interestingly and in contrast to respective empirical findings by Rao et al. (2023), classifier performance remains largely unaffected; Table 2 lists classifier F1 scores after pre-training (with only the classification loss) as well as after training with combined classification and segmentation loss.

Our unrolled ResNet50 architecture has 23.55M trainable parameters, whereas the respective UNet baseline has 38.84M. To make sure the inferior performance of the UNet is not just due to respective higher susceptibility to overfitting, we trained another UNet baseline based on a ResNet18 backbone, which has 18.81M parameters. However, the ResNet18 UNet is outperformed by the ResNet50 UNet, i.e., the fact that our UNet baseline has more parameters than our unrolled architecture does not explain its inferior performance.

Figures 2 and 3 show exemplary segmentation results, as well as the evolution of segmentations and heatmaps over training of unrolled LRP. We observe quick convergence to crisp segmentations, also in cases with multiple labels.

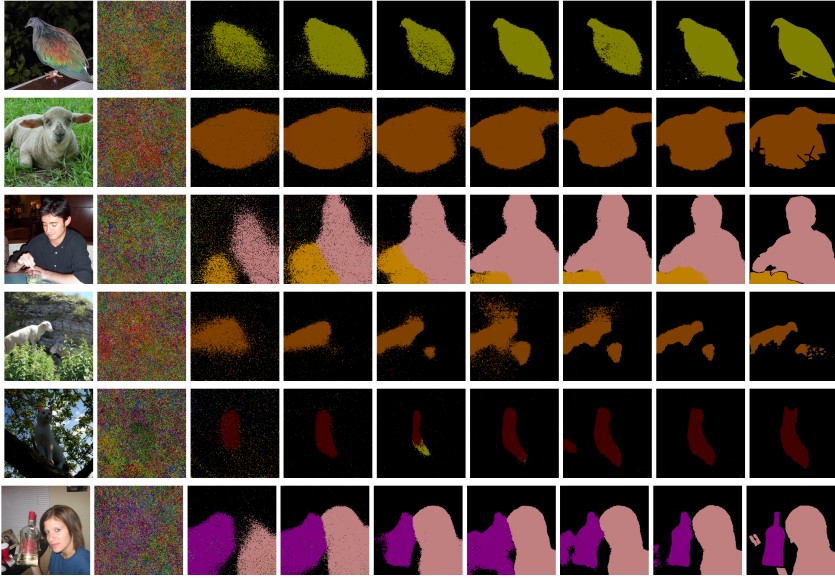

Figure 2: Exemplary unrolled LRP segmentations on the VOC validation set, 500 labelled images scenario. Left: input image; Right: ground truth segmentaion; In-between: Evolution of segmentation over training of unrolled LRP, ranging from the first iteration to peak validation mIoU.

| Method | tied ReLU | tied Conv | tied up-sample | concat skip | 1.4 % (20) | 6.8 % (100) | 34.2 % (500) | Full (1464) |
|---|---|---|---|---|---|---|---|---|
| Unrolled LRP-0 (ours) | ✓ | ✓ | ✓ | - | **39.80** | **50.29** | **58.30** | **61.50** |
| WS-AE | - | ✓ | ✓ | - | 29.50 | 46.59 | 54.74 | 57.91 |
| FCN | - | - | - | - | 26.23 | 39.98 | 51.27 | 56.64 |
| UNet | - | - | - | ✓ | 25.89 | 43.40 | 55.10 | 60.07 |

Table 3: Ablation of architectural elements between UNet and unrolled LRP. All results based on Resnet50 backbone; average mIoU on VOC val set over three independent training runs.

**Ablation Study.** Architectural elements that turn unrolled LRP into a UNet are: (1) Tied vs standard activation functions, (2) tied vs standard decoder weights and up-sampling, and (3) concat-skip connections from encoder to decoder. Table 3 lists results of a respective ablation study. The ablation includes a weight-sharing convolutional autoencoder (WS-AE), in which tied activations as well as the bottleneck heaviside function in unrolled LRP are replaced by standard ReLUs. WS-AE is trained with segmentation- and classification loss, like unrolled LRP. The ablation further includes a standard Fully Convolutional Network (FCN), in which weights in the decoder are free, and nearest neighbor upsampling is employed. To avoid the introduction of free weights per decoder branch, FCN is reduced to a single decoder branch. This entails that FCN does not contain the class weights of the classifier output layer. FCN is trained only with segmentaiton loss, like UNet.

**Discussion.** We hypothesize that superior performance of unrolled LRP over a comparable standard segmentation baseline in scenarios with limited supervision is due to the strong constraints imposed by tied weights, tied up-sampling, and tied activations. Furthermore, we hypothesize that the concordant losses as described in 2.3 facilitate the full exploitation of image-level labels for segmentation, as not just the segmentation loss, but also the classification loss pushes the total relevance in the heatmaps in a meaningful direction. As a quantitative indicator of this loss concordance, Table 2 reveals, to our knowledge for the first time, that vanilla classifiers can be trained in such a way that they condense all relevance to respective class foregrounds while classification performance remains unaffected. However, precisely how the sketched training dynamics play out for images with multiple class labels (including background) requires further study.

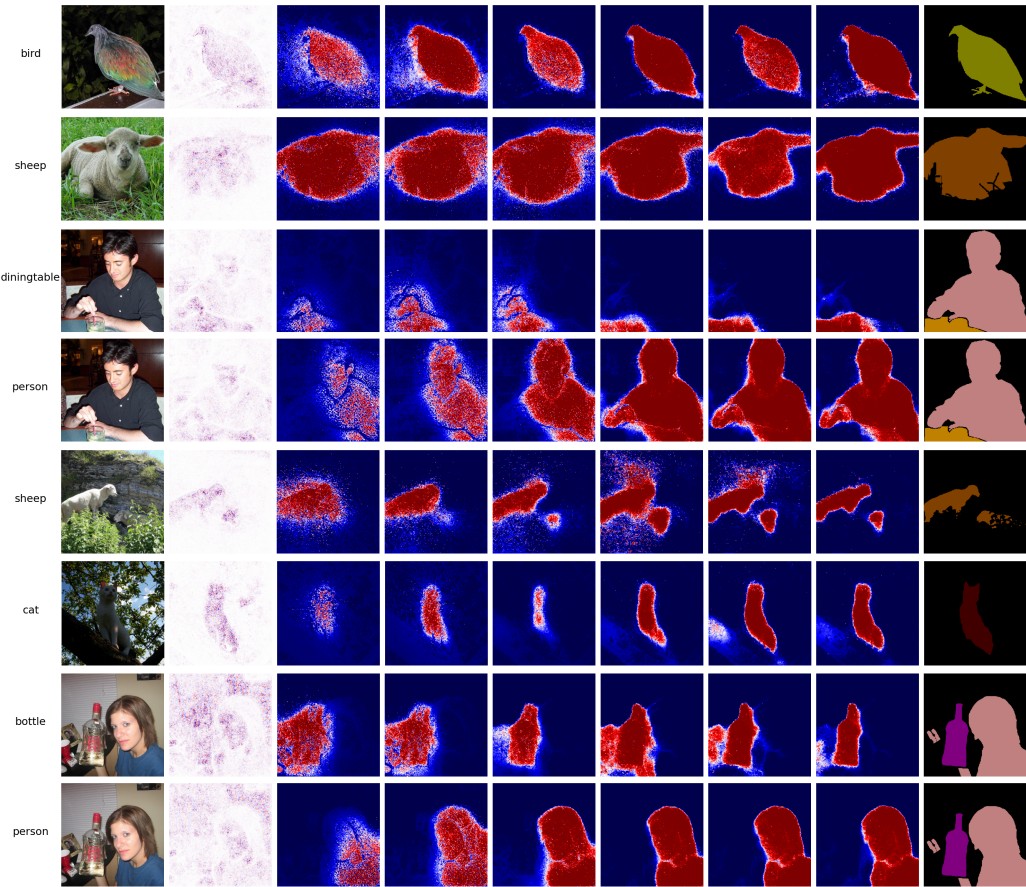

Figure 3: Exemplary unrolled LRP heatmaps on the VOC validation set, 500 labelled images scenario (cf. Fig. 2 for corresponding segmentations). Left: input image; Right: ground truth segmentaion; In-between: Evolution of heatmaps of the positive classes over training of unrolled LRP, ranging from the first iteration to peak validation mIoU. Note that all heatmaps except for the initial (left-most) one are shown after softmax to render standard blue-white-red visualization informative.

Our ablation study reveals that tied activation functions, an element our work newly introduces to segmentation architectures, are essential for the superior performance of LRP-0 over a standard UNet. Our study replicates previous findings that WS-AE outperforms FCN in few-labelled-samples regimes; yet it reveals that including tied activation functions yields another significant performance boost over WS-AE, which is crucial to outperforming the U-Net.

## 4 CONCLUSION

Our work establishes unrolled heatmap architectures as encoder-decoder-style convolutional architectures that can be trained for image segmentation. We observe superior performance to standard segmentation baselines in scenarios with limited pixel-level supervision, which entails the potential for practical use in semi-supervised segmentation. To this end, while not yielding state of the art semi-supervised segmentation results per se, our architecture directly lends itself to incorporation into orthogonal semi-supervised learning paradigms like augmentation consistency training. In contrast to previous work, we observe that training with a heatmap loss does not affect classification performance. This is striking in that our approach yields highly performant standard classifiers that nevertheless are trained to successfully focus on class foregrounds by means of our unrolled training objective.

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
