# OpenReview forum: "Model guidance via explanations turns image classifiers into segmentation models"
_ICLR.cc/2024/Conference — Submitted to ICLR 2024_

### Official Review · Reviewer_a33j · 2023-10-30

**Soundness:** 2 fair
**Presentation:** 1 poor
**Contribution:** 2 fair
**Rating:** 6
**Confidence:** 4

**Summary:**

The paper optimizes heatmaps towards improved segmentation performance. The authors establish formal parallels between differentiable heatmap architectures and conventional encoder-decoder architectures for image segmentation. Their experimental results demonstrate that unrolled LRP trained with combined classification and segmentation loss, can achieve competitive segmentation performance across comparable U-Nets. And the architectures they showcased exhibit favourable outcomes in certain weakly supervised training scenarios.

**Strengths:**

The paper establishes unrolled heatmap architectures as encoder-decoder-style convolutional architectures that can be trained for image segmentation.

The paper proposed the combined classification and segmentation loss and showed that differentiable heatmap architectures yield competitive results when trained with this loss.

The models proposed in this paper outperform comparable UNets in all supervision scenarios, with the performance margin increasing significantly as the level of pixel-level supervision decreases.

**Weaknesses:**

The paper did not conduct extensive datasets to verify the experimental results.

The contrast algorithm is restricted to U-Net and ignores relevant variants.

The paper specifically examines a limited number of cases comparing differentiable heatmap architectures with classical encoder-decoder architectures for image segmentation.

The language description is not clear, and there are some grammatical errors, such as case misuse.

**Questions:**

In Section 2.1 LRP BASICS of the paper, algorithm 1 should be presented with a clearer explanation.

In Section 2.2 UNROLLED LRP ARCHITECTURES FOR CONVOLUTIONAL CLASSIFIERS of the paper, what is the role of the 1x1 convolution mentioned in the "Final classifier layers" part?

In Section 2.3 LOSSES AND TRAINING of the paper, the different weight combinations of the loss function require more experimental support.

In Section 2.4 RELATION TO PREVIOUS FORMAL ANALYSES of the paper, “When trained with classification- and heatmap loss, the gradient of the classification loss backpropagates solely through the encoder, while the gradient of the heatmap loss backpropagates solely through the decoder. This can be leveraged for efficient training”, the phenomenon should be further validated through additional experiments or theoretical analysis in order to establish the credibility of this characteristic.

In Section 2.5 RELATION TO STANDARD ARCHITECTURES of the paper, unrolled heatmap architectures can unrolled heatmap architectures only be applied to U-Net or its related architectures? Can it be extended to a wider range of segmentation models? If not, please explain the reasons. If it can, please demonstrate its application and provide experimental comparisons with other architectures.

In Section 3 UNROLLED HEATMAP ARCHITECTURES FOR SEGMENTATION: RESULTS, please provide a more detailed description of the data selection and include validation on a wider range of datasets. Additionally, please include more comparative analysis regarding the improved U-Net models in the Quantitative Results part.

In Section 3 UNROLLED HEATMAP ARCHITECTURES FOR SEGMENTATION: RESULTS, “the ResNet18 UNet is outperformed by the ResNet50 UNet” ,the conclusion lacks experimental data support.

---

> ### Author Response · Authors · 2023-11-20
>
> Many thanks for your super helpful and detailed feedback.
>
> Regarding your questions:
>
> Re "In Section 2.1 LRP BASICS of the paper, algorithm 1 should be presented with a clearer explanation"\
> → We have revised Section 2.1 for clarity, please see updated manuscript
>
> Re "In Section 2.2 UNROLLED LRP ARCHITECTURES FOR CONVOLUTIONAL CLASSIFIERS of the paper, what is the role of the 1x1 convolution mentioned in the "Final classifier layers" part?" \
> → We have revised the “Final classifier layers” Paragraph of Section 2.2 for clarity, please see updated manuscript
>
> Re "In Section 2.3 LOSSES AND TRAINING of the paper, the different weight combinations of the loss function require more experimental support." \
> → Quantitative results from a loss weight hyperparameter tuning experiment are currently coming in; We will provide a respective update shortly.
>
> Re "In Section 2.4 RELATION TO PREVIOUS FORMAL ANALYSES of the paper, “When trained with classification- and heatmap loss, the gradient of the classification loss backpropagates solely through the encoder, while the gradient of the heatmap loss backpropagates solely through the decoder. This can be leveraged for efficient training”, the phenomenon should be further validated through additional experiments or theoretical analysis in order to establish the credibility of this characteristic." \
> → The requested theoretical analysis is comprehensively performed by Etman et al. (2019); We had added this reference in the preceding paragraph, but will add it again next to the above statement again to clarify that this has been known and studied before
>
> Re "In Section 2.5 RELATION TO STANDARD ARCHITECTURES of the paper, unrolled heatmap architectures can unrolled heatmap architectures only be applied to U-Net or its related architectures? Can it be extended to a wider range of segmentation models? If not, please explain the reasons. If it can, please demonstrate its application and provide experimental comparisons with other architectures." \
> → As LRP-0 works layer-by-layer, extension of unrolled heatmap architectures to pyramid networks appears straightforward. We will add this to our discussion of possible future directions.
>
> Re "In Section 3 UNROLLED HEATMAP ARCHITECTURES FOR SEGMENTATION: RESULTS, please provide a more detailed description of the data selection " \
> → random selection of labelled images, with fixed seed across all experiments
>
> Re "..and include validation on a wider range of datasets. " \
> → We are working on extending our validation to CityScapes, yet due to the limited compute we have at hand results will unfortunately not come in during the Discussion Phase
>
> Re "Additionally, please include more comparative analysis regarding the improved U-Net models in the Quantitative Results part." \
> → We’ve added an ablation study in which we evaluate known architectures “in between” unrolled LRP-0 and U-Net, namely a weight-sharing convolutional autoencoder (WS-AE) and the well-known Full Convolutional Network (FCN), see Table 3 in the updated manuscript. This reveals that tied activation functions, an element our work newly introduces to segmentation architectures, are essential for the superior performance of LRP-0. Our study replicates previous findings that WS-AE outperforms FCN in few-labelled-samples regimes – Yet it reveals that including tied activation functions yields another significant performance boost over WS-AE, which is crucial to outperforming the U-Net.
>
> Re "In Section 3 UNROLLED HEATMAP ARCHITECTURES FOR SEGMENTATION: RESULTS, “the ResNet18 UNet is outperformed by the ResNet50 UNet” ,the conclusion lacks experimental data support." \
> → We will include ResNet18 UNet results in the Supplement (Results are, for 20, 100, 500, 1464 labelled images: mIoU 18.46, 33.95, 49.40, 54.35 respectively)

---

> > ### Author Response · Authors · 2023-11-22
> >
> > Here's the quantitative results from our loss weight hyperparameter tuning experiment as promised; all on VOC val set in full labelled image scenario, all with classification loss weight 1:
> >
> > | segmentation loss weight | 2e-4 | 2e-3 | 0.02 | 0.2 | 1.0 | 2.0 | 5.0 |
> > | ----------- | ----------- | ----------- | ----------- | ----------- | ----------- | ----------- | ----------- |
> > | classification F1 | 82.33 | 82.17 | 82.56 | 81.00 | 81.49 | 81.10 | 81.16 |
> > | segmentation mIoU | 60.73 | 61.29 | 61.10 | 60.69 | 61.50 | 60.57 | 61.06 |
> >
> > Results show that classification performance is largely unaffected even when cranking up the segmentation loss weight way beyond what is necessary to achieve peak segmentation performance.

---

### Official Review · Reviewer_K7Hs · 2023-11-02

**Soundness:** 2 fair
**Presentation:** 1 poor
**Contribution:** 2 fair
**Rating:** 3
**Confidence:** 4

**Summary:**

The paper presents a method to turn image classifiers into segmentation models.

**Strengths:**

It is a promising research direction to turn image classifiers directly into segmentation models, especially considering that there are many well-performed pre-trained (vision-language) classification models.

**Weaknesses:**

1. The paper lacks important comparisons with other weakly-supervised semantic segmentation methods that can also extract pseudo semantic masks from image-level labels.
2. The authors claim they do not want to achieve the best semi-supervised performance, but the reported results are unacceptably too poor. And there seems not to be any ablations studies on the proposed method. It is strongly recommended to re-prepare the draft. I do not think this work has been well prepared for ICLR submission.

**Questions:**

Please refer to the above weaknesses.

---

> ### Author Response · Authors · 2023-11-20
>
> Many thanks for your feedback.
>
> Re “Lack of comparison to other weakly-supervised methods that work with image-level labels:” \
> So far we had summarized related weakly-supervised segmentation methods only very briefly in the “Limitations” Section of the Introduction. We will provide an extended discussion in an additional paragraph in the “Related Works” Section of the Introduction, mentioning that most existing methods (other than the Li et al. (2018), which is already discussed in detail) are based on vanilla Grad-CAM followed by sophisticated post-processing as opposed to our heatmap loss-based training.
>
> Re “Lack of ablation studies”: \
> We’ve added an ablation study in which we evaluate known architectures “in-between” unrolled LRP-0 and U-Net, namely a weight-sharing convolutional autoencoder (WS-AE) and the well-known Full Convolutional Network (FCN), see Table 3 in the updated manuscript. This reveals that tied activation functions, an element our work newly introduces to segmentation architectures, are essential for the superior performance of LRP-0. Our study replicates previous findings that WS-AE outperforms FCN in few-labelled-samples regimes – Yet it reveals that including tied activation functions yields another significant performance boost over WS-AE, which is crucial to outperforming the U-Net.

---

### Official Review · Reviewer_43DH · 2023-11-06

**Soundness:** 4 excellent
**Presentation:** 3 good
**Contribution:** 3 good
**Rating:** 8
**Confidence:** 4

**Summary:**

This paper establishes formal parallels between differentiable heatmap architectures and conventional encoder-decoder architectures used for image segmentation. It conducts a comparative evaluation of these two approaches in terms of segmentation accuracy, finding that differentiable heatmap architectures, when trained with combined classification and segmentation loss, can achieve competitive segmentation performance. The authors also explore semi-supervised training with varying numbers of pixel-level labels, showing that differentiable heatmap architectures outperform standard U-Nets for segmentation in scenarios with few pixel-wise labels.

**Strengths:**

- This paper is well written and structured. I enjoyed reading this paper and find the idea quite interesting. The proposed unrolled LRP for benefits from having layer-wise guidance from the heatmaps with the skip and tied-in connections for accurate segmentation map prediction. However, it remains somewhat unclear whether this directly leads to improved segmentation performance, as indicated by the authors. It might be the case that the segmentation results are more closely tied to the quality of the heatmaps. Nevertheless, I feel that the idea is novel and is of sufficient interest to the research community.
- The paper makes use of standard training objectives with the proposed unrolled LRP for semi-supervised segmentation, thereby making it directly usable with any task specific architectures. In general, I am in favour of simple and easy to plug-in methods that can complement already existing approaches.
- The proposed method is well supported by experiment results. The empirical finding that concurrent training for classification and segmentation does not compromise classifier performance and holds up comparably to conventional segmentation architectures is quite intriguing. This finding suggests that the method is more generalizable to classification as well as segmentation.

**Weaknesses:**

- Continuining from one of my speculations I mentioned in strengths (point 1), the segmentation performance might be closely tied to the quality of heatmaps. This heatmap quality significantly based on amount of available data and the distribution of classes within a dataset. We already know that classification approaches tend to suffer in performance when there is imbalance in the number of samples per class, and I suspect that challenge may also extend to the segmentation performance. How robust is this method to such real-world scenarios with dataset and class imbalances?

**Questions:**

- Strengths (point 1) and weaknesses sections have an unanswered question for the authors to respond. I have listed a few more questions below.
- Subsequently, bigger multi-label datasets with large number multiple instances per image can also create more uncertain regions the heatmaps. Is the method able to handle this?
- How would the losses from orthogonal semi-supervised segmentation approaches affect the training with an unrolled LRP? Do you expect to see better performances?
- There is no code currently available, will the authors make it available at some point?

---

> ### Author Response · Authors · 2023-11-20
>
> Many thanks for your highly encouraging assessment and very helpful feedback.
>
> Regarding your questions:
>
> Re “Robustness to class imbalance / heatmap quality:” \
> Inhowfar segmentation performance depends on initial heatmap quality is a very interesting question. As you state, multiple factors contribute to heatmap quality, among which (1) class imbalance, (2) many diverse semantic classes and/or instances per image, and (3) confounders (with Clever-Hans in ImageNet among the most well-studied). The “Right for the Right Reason” community has extensively shown that issue (3) can be mitigated via losses on heatmaps, and thus we hypothesize this to be the case also for our segmentation loss. However, (1) and (2) also affect the performance of standard segmentation networks. Here, a range of methodology has been proposed to mitigate this effect, e.g. via image- and pixel sampling strategies tailored to counter detrimental effects of high class imbalance. Consequently, to us, an extremely interesting question for further study appears to be: Are there circumstances under which standard segmentation networks (together with their mitigation methodology) *are* able to handle a challenging situation, whereas unrolled heatmap segmentation architectures (with same mitigation methodology) *are not*? So far, vaguely speaking, our results suggest that unrolled LRP behaves “similar enough” to UNets that we would not expect any “surprises” regarding inferior performance in specific situations. E.g., we expect class imbalance treatment as e.g. in Araslanov et al. (2021) to have a similar-sized positive effect no matter if standard- or unrolled heatmap segmentation architectures are employed. However, substantiating this hypothesis via further studies is a very interesting direction, and we will add a respective discussion to the Discussion Section of the paper.
>
> Re “Would the losses from orthogonal semi-supervised segmentation approaches affect the training with an unrolled LRP?” \
> Along the same lines of reasoning as above, we expect improvements by means of orthogonal semi-supervised approaches (like e.g. augmentation consistency training) to be of sizes comparable to respective improvements for standard segmentation networks. As above, this hypothesis needs to be substantiated in further studies.
>
> Re “There is no code currently available, will the authors make it available at some point?” \
> Yes, all code will be made available upon publication of the manuscript.

---

### Official Review · Reviewer_nMm3 · 2023-11-06

**Soundness:** 3 good
**Presentation:** 1 poor
**Contribution:** 2 fair
**Rating:** 5
**Confidence:** 4

**Summary:**

This paper studies the weak supervision with image-level labels to achieve segmentation. It establishes formal parallels between differentiable heatmap architectures and conventional encoder-decoder architectures commonly used for image segmentation.

**Strengths:**

The studied weak form supervision is interesting and helps understanding the learning of convolution neural networks.

**Weaknesses:**

The organization and presentation of this paper is poor, and some writing and language use is vague, the paper also lacks clear presentation of its contribution in the context of prior research work. for example, the LRP is used many times in the abstract, main text, image caption, section title, e.t.c, but without given concrete definition, which makes the paper quality poor.

**Questions:**

N.A.

---

> ### Author Response · Authors · 2023-11-20
>
> Many thanks for your helpful feedback.
>
> Re “Contribution”: \
> Our main contribution is that we are first to report formal parallels between unrolled heatmap architectures and standard segmentation networks. We showcase the respective potential of unrolled heatmap architectures by empirical results on Pascal VOC, where they outperform standard segmentation baselines.
>
> Re “Clarity of writing”: \
> We have revised large parts of the Methods Section for clarity. Please see updated manuscript; Revised Paragraphs are highlighted.

---

### Meta-Review · Area_Chair_1WCU · 2023-12-05

**Metareview:**

This work proposes a unified approach for generating heat maps for explainability and also for weakly-supervised semantic segmentation (WSSS). All reviewers agree that the proposed method is somewhat novel, and interesting. The main initial concerns were (i) lack of clarity, e.g. of LRP; and (ii) lack of comprehensive comparisons or baselines (e.g. restriction to U-Net architectures, no meaningful comparison to SOTA methods for explainability or WSSS). While the authors claim that achieving SOTA WSSS results is not the aim of the work, some additional quantitative and qualitative comparison is needed to give some sense of how close the proposed method is to SOTA methods, especially if the work claims to perform well on both tasks. The authors addressed concern (i) during the discussion period. However, there were still concerns about (i) the lack of a more detailed discussion of related work and (ii) no quantitative comparison to any SOTA WSSS or explainability works. The final reviewer ratings were mixed. The AC agrees with the recommendation to not accept for reason (ii), and recommends that the authors include a more robust analysis and evaluation section in a future version of this work. In particular, AC recommends to include at least some comparisons to related WSSS and explainability works without post processing.

**Justification For Why Not Higher Score:**

See above

**Justification For Why Not Lower Score:**

n/a

---

### Decision · Program_Chairs · 2024-01-16

Reject